Subject Area:
molecular biology/cellular biology/biotechnology

Keywords:
RNA interference, double-stranded RNA, liposome, melanin

Author for correspondence:
Steve Whyard
e-mail: steve.whyard@umanitoba.ca

# Efficiency of RNA interference is improved by knockdown of dsRNA nucleases in tephritid fruit flies

Alison Tayler, Daniel Heschuk, David Giesbrecht, Jae Yeon Park and Steve Whyard

Department of Biological Sciences, University of Manitoba, Winnipeg, Manitoba, Canada R3T 2N2

AT, 0000-0002-0510-1551; DG, 0000-0003-2498-4705; SW, 0000-0002-9874-7780

RNA interference (RNAi) in insects is routinely used to ascertain gene function, but also has potential as a technology to control pest species. For some insects, such as beetles, ingestion of small quantities of double-stranded RNA (dsRNA) is able to knock down a targeted gene's expression. However, in other species, ingestion of dsRNA can be ineffective owing to the presence of nucleases within the gut, which degrade dsRNA before it reaches target cells. In this study, we observed that nucleases within the gut of the Queensland fruit fly (*Bactrocera tryoni*) rapidly degrade dsRNA and reduce RNAi efficacy. By complexing dsRNA with liposomes within the adult insect's diet, RNAi-mediated knockdown of a melanin synthesis gene, *yellow*, was improved significantly, resulting in strong RNAi phenotypes. RNAi efficiency was also enhanced by feeding both larvae and adults for several days on dsRNAs that targeted two different dsRNase gene transcripts. Co-delivery of both dsRNase-specific dsRNAs and *yellow* dsRNA resulted in almost complete knockdown of the *yellow* transcripts. These findings show that the use of liposomes or co-feeding of nuclease-specific dsRNAs significantly improves RNAi inhibition of gene expression in *B. tryoni* and could be a useful strategy to improve RNAi-based control in other insect species.

## 1. Introduction

RNA interference (RNAi) has become a widely used reverse genetics tool in insects, owing largely to the relative ease of knocking down a targeted gene's transcripts simply by injecting double-stranded RNA (dsRNA) into the insect's haemocoel (reviewed in [1]). In more recent years, the focus of RNAi applications in insects has turned to the development of new methods of pest insect control. Because of its sequence specificity, RNAi has the potential to provide a new generation of species-specific pesticides that target transcripts within a pest species, but do not adversely affect beneficial or non-target species [2–4]. To this end, transgenic plants expressing insecticidal dsRNAs have been produced that provide effective control against specific pests [5,6] and foliar dsRNA sprays have also proven effective in controlling herbivorous insect pests in laboratory trials [7]. In addition to these insecticidal applications, dsRNAs that target genes involved in male fertility and/or female development have been administered to pest insects with the aim of producing populations of sterile males for sterile insect technique (SIT) applications [8–10].

For large-scale uses of RNAi such as foliar sprays or SIT applications, the most effective delivery method of dsRNAs to insects is through feeding. For many coleopterans, ingested dsRNA has proven highly successful as a method to induce potent and systemic RNAi (reviewed in [11]). For other insects, the efficacy of ingested RNAi can be variable. Nucleases capable of degrading dsRNAs have been detected in the gut, saliva and haemolymph in various insects [12–16] and these have been implicated for the failure to achieve

efficient RNAi in some species. For example, in many lepidopteran species, RNAi is particularly ineffective, largely because of the presence of nucleases in both the gut and the haemolymph [17,18]. Evidence supporting that nucleases can have significant impacts on RNAi efficacy has been confirmed by reducing nuclease activity by delivering nuclease-specific dsRNA to the insects. In the locust *Locusta migratoria*, knockdown of a gut dsRNase by haemocoel injections of nuclease-specific dsRNA resulted in considerably improved RNAi [19]. Even insects with relatively good RNAi responses to ingested dsRNAs, such as the Colorado potato beetle, *Leptinotarsa decemlineata*, showed improved RNAi efficacy when nuclease gene transcripts were knocked down [20]. In that study, the authors observed improved RNAi-mediated knockdown of target gene transcripts in the beetles by first feeding the insects dsRNA targeting two gut nucleases. Similarly, Chung *et al.* [21] induced knockdown of bacterial symbiosis genes in the pea aphid *Acyrthosiphon pisum* by co-feeding a mixture of dsRNAs targeting both symbiosis-related genes and a gut nuclease gene. In these cases, enough nuclease-specific dsRNAs are thought to have entered the gut cells of the insects in the early feedings, reducing nuclease activity in the gut and thereby improving the efficacy of RNAi within a few days of further dsRNA consumption.

The applicability of delivering dsRNA targeting gut nucleases to improve RNAi efficiency in other insects has not been thoroughly investigated. In one of our previous studies, we observed that feeding dsRNA to adult Queensland fruit flies could induce RNAi, but we failed to induce RNAi in larvae [10]. The Queensland fruit fly, *Bactrocera tryoni* (Diptera: Tephritidae), is Australia's most economically damaging pest insect, currently threatening much of eastern Australia's horticulture crops [22]. Several pest management strategies, including organophosphate insecticides, bait-sprays and parasitoid wasp releases, have been used to control this pest [23–25]. SIT, which involves the release of large numbers of sterile males to suppress mating efficiency in the field, has also been used for decades as a species-specific control strategy for this pest [26]. Sterilization of males in SIT programmes is typically accomplished using low-dose radiation, which can, in some insects, reduce the mating competiveness of the released males [26]. As an alternative to radiation, we and others have considered using RNAi to produce sterile males [8,10,27]. In our previous study with *B. tryoni*, we observed that RNAi efficiency was suboptimal, and we recognized that improvements would be required before the technology could be considered as a viable and cost-effective alternative to conventional sterilization methods.

In this study, we observed that dsRNA was rapidly degraded by nucleases within the gut of *B. tryoni*. Two gut nucleases were identified, and by feeding the insects nuclease-specific dsRNAs the efficacy of RNAi targeting other genes' transcripts was considerably improved. For this particular study, we selected a target gene encoding melanin biosynthesis, which when knocked down by RNAi resulted in a measurable, yet non-lethal phenotype. By simultaneously feeding insects both nuclease-specific and other dsRNAs, or by encapsulating dsRNAs in liposomes within the diet, we observed large reductions in the target gene transcripts and potent phenotypes. This co-feeding delivery method proved effective for both adult and larval stages and could prove

useful for other insects that show limited RNAi efficiency with dsRNA-degrading nucleases.

# 2. Material and methods

## 2.1. Insect culture

*Bactrocera tryoni* were kindly provided by Dr Solomon Balagawi (Elizabeth Macarthur Agriculture Institute, Australia), and were derived from wild flies reared from fruits collected in 2013 from Griffith and Gosford, NSW, Australia. Adult flies were reared at 28°C, 75% relative humidity with a photoperiod of 14:10 h (light:dark) and were provided sugar cubes and water. A torula yeast paste was also provided to promote egg development. Eggs were laid on apple skins and transferred to a carrot-based artificial medium [28]. Wandering larvae were transferred to Petri dishes with autoclaved sand to allow larvae to pupate, and pupae were then transferred to the colony cages (30 cm × 30 cm × 30 cm) or placed in individual cotton-stoppered vials (25 ml) for treatment [10].

## 2.2. Phylogenetic analyses

The putative nuclease gene transcript sequences, *dsRNase1* and *dsRNase2*, as well as the putative *yellow* gene were identified using nucleotide BLAST (NCBI) from predicted gene-coding sequences in *Bactrocera dorsalis* and a full *B. tryoni* genome assembly search (GCA_000695345.1, NCBI). To generate phylogenetic trees, full-length sequences were aligned in Clustal Omega and trimmed to include only aligned sequences. Neighbour-joining trees were generated and tested in MEGA X.

## 2.3. Preparation of dsRNA

DsRNAs targeting *dsRNase1*, *dsRNase2* and two non-*B. tryoni* control genes, *green fluorescent protein* (*gfp*) and *β-glucuronidase* (*gus*), were purchased from AgroRNA (Seoul, South Korea) using the DNA sequences in electronic supplementary material, table S1, and were purified using standard desalting procedures. To synthesize dsRNAs targeting *yellow*, gene-specific polymerase chain reaction (PCR) primers containing the restriction sites *XhoI* and *XbaI* were designed to amplify a 347 bp fragment (electronic supplementary material, table S2). The PCR products were digested using *XbaI* and *XhoI* restriction enzymes and ligated into the similarly digested plasmid pL4440, a vector possessing convergent T7 promoters. DNA templates for *in vitro* transcription of each of the gene fragments in pL4440 were PCR-amplified using the following pL4440-specific primers: pL4440F (ACCTGGCTTATCGAA) and pL4440R (TAAAACGACGGC-CAGT). PCR products were purified using a gel extraction kit (GeneJET, Thermo Fisher Scientific). The MEGAscript RNAi kit (Ambion) was then used for *in vitro* transcription and purification of dsRNAs following the manufacturer's protocol.

## 2.4. *Ex vivo* nuclease activity assays

Flies were dissected in phosphate-buffered saline (PBS) to collect the gut from mouth to anus, excluding the crop and Malpighian tubules. Guts were initially homogenized,

royalsocietypublishing.org/journal/rsob Open Biol. 9: 190198

sliced into six tubular pieces or left intact with or without the food bolus to determine whether nuclease activity was detectable in the various gut treatments. As no differences were observed in nuclease activity using each tissue preparation method, all subsequent *ex vivo* assays were performed with guts sliced into tubular pieces. Three guts were pooled into 0.6 ml tubes containing 100 µl of PBS and refrigerated at 4°C for 16 h to allow enzymes to dissipate into the solution. The tissues and liquid were subjected to centrifugation at 13 000$g$ for 5 min. The supernatant was then collected and diluted 1 : 2 with molecular grade water. Then, 750 ng of *gus*-dsRNA was added to 30 µl of the diluted solution, either alone or in combination with 1.125 µl of Lipofectamine™ 3000 and 1.5 µl of P3000™. Control gut samples were heat-inactivated at 65°C for 30 min prior to adding the dsRNA. The dsRNA–gut mixture was incubated at 28°C and 10 µl aliquots were removed after 10 and 60 min. Samples were kept on ice until visualization using 1.5% agarose gel electrophoresis. Band fluorescence intensities were measured using Image Lab™ 6.0 software (BioRad).

## 2.5. dsRNA delivery to flies

Ten newly eclosed adult flies were microinjected in the dorsal thorax with 1.0 µl of *dsRNase1*, *dsRNase2* or *gfp* dsRNA as a control, at a concentration of 1.0 µg µl$^{-1}$, using glass borosilicate needles (Fisher) and a FemtoJet microinjector (Eppendorf). For the *dsRNase1 + dsRNase2* dsRNA combination treatments, 0.5 µl of each dsRNA, totalling 1 µl, was injected into each adult. The doses chosen were similar to those used to produce an effective transcript knockdown in a closely related species, *B. dorsalis* [29,30]. Flies were kept in individual cotton-stoppered plastic vials (25 ml) and fed droplets of a 10% sucrose solution until the time of dissection 6 days later.

To measure RNAi efficiency following injection of dsRNA against nucleases, adult flies were first injected with *gfp* or a combination of *dsRNase1* and *dsRNase2* dsRNA as described above. Flies were then fed daily with a dose of 2 µg of *gfp*- or *yellow*-dsRNA dissolved in 10 µl of a 10% sucrose solution for 6 consecutive days. The dsRNA-containing sucrose droplet was added to the bottom of the vials daily and flies consumed the entire droplet in that time. Control vials containing no flies confirmed that the droplets did not evaporate over that time period. A second treatment involved feeding adults 2 µg of dsRNA targeting *gfp* for three consecutive days followed by 2 µg of *yellow*-dsRNA for an additional 3 days, or 2 µg each of *dsRNase1* and *dsRNase2* for 3 days followed by 2 µg of *yellow*-dsRNA for an additional 3 days. In a third treatment, adults were fed 2 µg of *gfp*-dsRNA or *yellow*-dsRNA or co-fed 2 µg of both the nuclease-specific- and 2 µg of *yellow*-dsRNA for 6 consecutive days. In a fourth treatment, flies were fed a combination of 2 µg of *gfp*-dsRNA or *yellow*-dsRNA, 3 µl of Lipofectamine 3000 and 4 µl of P3000 reagent dissolved in 10 µl of a 10% sucrose solution daily for 6 consecutive days. To assess for RNAi, flies were collected on the final day of dsRNA treatment and RNA was extracted from whole bodies for subsequent quantitative reverse-transcriptase PCR (qRT-PCR) analysis.

For feeding of dsRNA to larvae, eggs were hatched in 35 mm Petri dishes containing either 50 µl of dsRNA at a concentration of 1 µg µl$^{-1}$ or a dsRNA-liposome mixture containing 50 µl of dsRNA (1.0 µg µl$^{-1}$), 75 µl of Lipofectamine

3000 and 100 µl of P3000 reagent. For dsRNA combination treatments of *gfp + yellow* dsRNA or *dsRNase1 + dsRNase2 + yellow* dsRNA, 50 µl of each dsRNA was mixed and added to the dish at a concentration of 1.0 µg µl$^{-1}$. Twenty-four hours later, groups of five first instar larvae were collected from the dish and placed together into individual wells of 24-well plates. A carrot-based artificial medium (0.25 g) [28] was placed in each well and dsRNA was added to the medium at a dose of 10 µg per well (1 µg per larva), either alone or mixed with 15 µl of Lipofectamine 3000 and 20 µl of P3000 reagent. For the combination treatments of *gfp + yellow* dsRNA and *dsRNase1 + dsRNase2 + yellow* dsRNA, 5 µl of a 1.0 µg µl$^{-1}$ solution (1 µg per larva) of each dsRNA was mixed and added to the wells. Larvae were transferred to fresh food containing dsRNA each day. After 4 days, the volume of dsRNA added to the food increased to 10 µl (2 µg per larva) and the amount of medium increased to 0.5 g to accommodate the growing size of the larvae. After 7 consecutive days of treatment, larvae were removed from the wells and placed into 35 mm Petri dishes lined with 5 g of medium. The Petri dishes were placed into plastic containers lined with autoclaved sand to allow the larvae to pupate. Adults that eclosed were collected into individual cotton-stoppered vials, fed on a 10% sucrose solution diet and allowed to develop for 3, 7 or 10 days before being sacrificed for RNA extractions.

## 2.6. Melanization assays

Newly eclosed adult female flies were fed 2 µg of *gfp*- or *yellow*-dsRNA, either naked or in combination with Lipofectamine 3000, for 6 consecutive days as described above. Haemolymph was extracted from anaesthetized flies by removing one of the posterior legs and using a glass capillary micropipette to collect approximately 0.5 µl haemolymph from the wound. Pooled haemolymph from 10 flies was mixed together and total protein was measured using a spectrophotometer at 280 nm. Haemolymph was diluted with PBS to a protein concentration of 3.5 µg µl$^{-1}$. Melanization was measured by mixing 1 µl of diluted haemolymph with 30 µl of PBS and 3 µl of a 1 mg ml$^{-1}$ solution of the microbial elicitor curdlan, a β-1,3-glucan polymer, to promote melanization [31]. The production of melanin was measured at 28°C using a BioTek Synergy H1 plate reader, reading the absorbance at 470 nm every 15 min. Six independent biological replicates were collected for each treatment.

## 2.7. RNA isolation

Gut tissue from treated insects was dissected in PBS from both larvae and adults and haemolymph was extracted from adults using 1 mm borosilicate glass needles (pulled using a Flaming-Brown Micropipette Puller; Sutter Instruments Co., Novata, CA, USA). Haemolymph from 10 pooled adults, guts from three pooled adults and single whole bodies were placed directly in 100 µl of lysis buffer supplemented with 2% β-mercaptoethanol and stored at −80°C until further use. Total RNA was extracted from these tissues using QIAshredder (Qiagen) columns to homogenize tissues and a GeneJET RNA purification kit (Thermo Fisher Scientific). Contaminating genomic DNA was removed using an RNase-free DNase I (Thermo Fisher Scientific) treatment. RNA was quantified and purity was assessed using a Biochrom NanoVue UV–Vis

royalsocietypublishing.org/journal/rsob    Open Biol. **9**: 190198

spectrophotometer. Complementary DNA (cDNA) was synthesized with a qScript cDNA Supermix kit (Quanta Biosciences). The purity of the cDNA was verified by PCR amplification using a Lucigen EconoTaq PLUS 2X Master Mix (following the manufacturer's protocol) with *actin-2*-specific primers (electronic supplementary material, table S2), and subsequent 1.5% agarose gel electrophoresis.

## 2.8. Quantitative RT-PCR analysis

Quantitative RT-PCR was used to determine the expression patterns of genes within *B. tryoni* tissues, and to assess impacts of the dsRNA treatments. The qRT-PCR reactions were performed using primers designed from sequences acquired from the assembled *B. tryoni* genome database (http://www.ncbi.nlm.nih.gov/genome/15403) (electronic supplementary material, table S2). As an internal control, a fragment of the *actin-2* gene (NCBI reference sequence: L12255, hereafter referred to simply as *actin*) was also amplified to normalize the amount of cDNA added to the qRT-PCR reactions. This reference gene was chosen because it is expressed in similar levels among tissues and developmental stages, and has been used as a standard in other insect species [16,32]. It is also the preferential reference gene used to assess gene expression in gut tissue in the closely related species *B. dorsalis* [33]. A single reference gene was considered sufficient, as the primer efficiencies of *actin* among tissues and developmental stages were within 1.5% of one another (96–97.5%), and primer efficiencies for all other genes ranged within 2.5% of *actin* within a given tissue. Quantitative RT-PCR amplifications were performed with SsoFast Evagreen Supermix (BioRad) according to the manufacturer's specifications using the BioRad CFX Connect Real-Time PCR System. Melting curve analyses were performed to ensure specificity and consistency of all PCR-generated products. All reactions were repeated in duplicate (technical replicates) and 10 biological replicates were analysed for each gene targeted with dsRNA to examine tissue and stage specificity. Quantification of the transcript level was performed according to the $2^{-\Delta\Delta CT}$ value method [34].

## 2.9. Statistical analysis

Significant differences between treatments and controls in the nuclease and *yellow* marker gene knockdown trials and in the dsRNA degradation assays were evaluated using a Welch's *t*-test for two independent sample groups. Statistical analyses were performed using the Prism (GraphPad) software with a significance level of 0.05. Normality and homogeneity of variances were tested using the Kolmogorov–Smirnov test and Levene's test, respectively. In cases where variables did not meet the normality and/or homogeneity premises, they were $log_{10}$ transformed. Statistical analyses were performed in the Statistica 10.0 software (Dell Software, Round Rock, TX, USA) with a significance level of 0.05.

# 3. Results

## 3.1. Nuclease gene identification and tissue specificity

Two putative nuclease genes (*dsRNase1* and *dsRNase2*) were identified within the *B. tryoni* genome, with each having

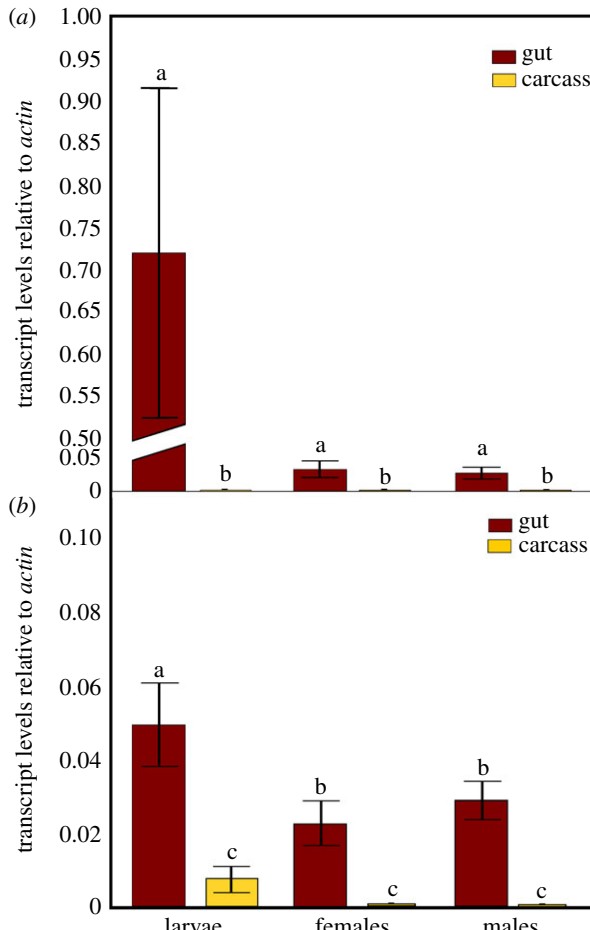

**Figure 1.** Transcript levels of (*a*) *dsRNase1* and (*b*) *dsRNase2* genes, relative to *actin*, in gut or carcass of larvae or adults (males and females) of *B. tryoni*. Values represent the means and standard errors of three biological replicates; different letters indicate significant differences using ANOVA with Tukey's *post hoc* test, $p < 0.05$.

greater than 67% nucleotide sequence identity to two *Drosophila melanogaster* genes predicted to be nucleases (GO term 0004519) with high expression in the gut. The *B. tryoni* genes were both over 94% identical to two *B. dorsalis* genes also predicted to be nucleases (electronic supplementary material, table S3). We included these predicted dipteran nuclease genes along with confirmed dsRNase genes identified in a range of other insects in a phylogenetic analysis and found the *B. tryoni* genes are less related (51% nucleotide identity) to each other and more like different nuclease genes of other insects (electronic supplementary material, figure S1). Quantitative RT-PCR analyses determined that *dsRNase1* was expressed exclusively within the gut of larval and adult male and female *B. tryoni* (figure 1*a*) while *dsRNase2* was expressed exclusively in the guts of adults but was also expressed in other tissues within larvae (figure 1*b*).

## 3.2. Suppression of gut nucleases using RNAi

Nuclease activity in excised adult insect guts was assessed in *ex vivo* assays. Regardless of whether the dissected guts were homogenized, sliced into 1–2 mm length pieces or left largely intact, nuclease activity was readily detected by mixing the gut extract or dissection medium with *gus*-dsRNA. For all subsequent analyses, we used only sliced guts to minimize potential effects from intracellular nucleases. The *gus*-dsRNAs were resolved on agarose gels,

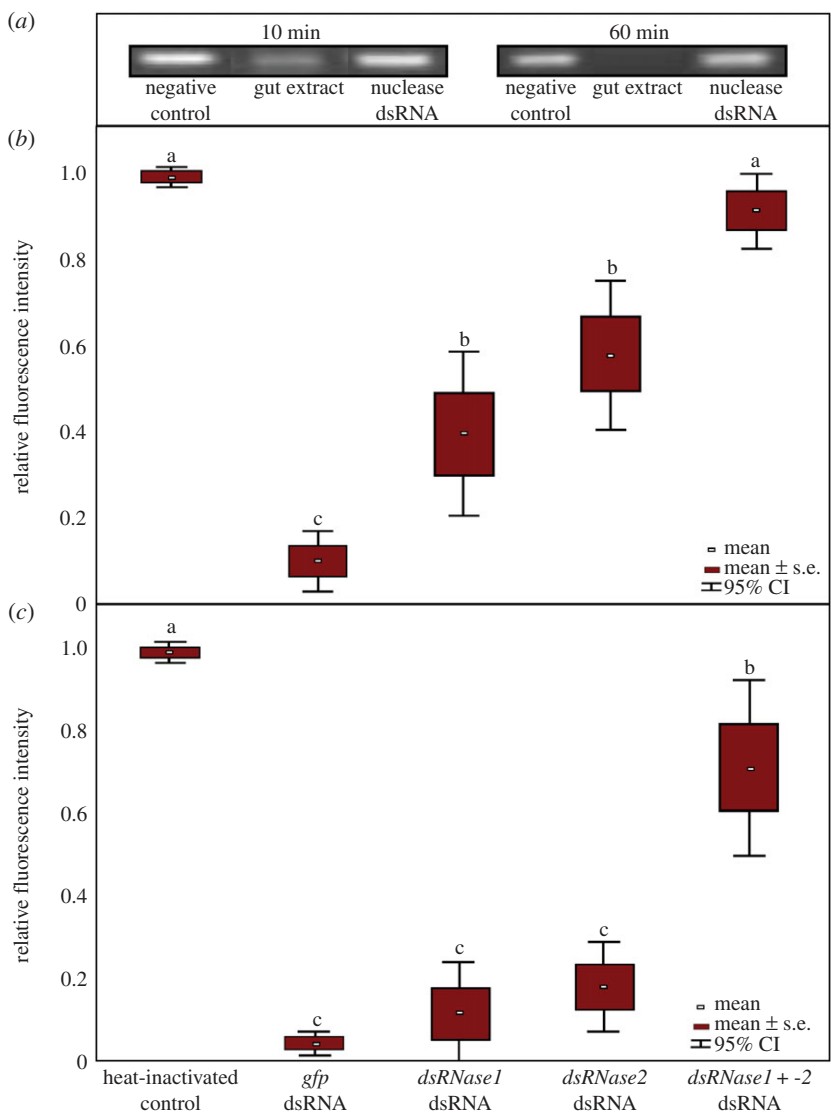

royalsocietypublishing.org/journal/rsob    Open Biol. **9**: 190198

**Figure 2.** Degradation of *gus*-dsRNA incubated in *B. tryoni* gut contents in adults injected with 1 μg of dsRNA targeting the gut-specific nucleases *dsRNase1* and *dsRNase2*. DsRNAs exposed to gut contents were resolved on agarose gels and Image Lab software was used to determine band fluorescence intensity. (*a*) Representative gel images of dsRNA at 10 and 60 min exposure to gut extracts. Lane 1 = *gus*-dsRNA not exposed to gut extracts; lane 2 = *gus*-dsRNA exposed to gut extracts, showing considerable or complete degradation at 10 and 60 min, respectively; lane 3 = *gus*-dsRNA exposed to gut extracts of flies injected 24 h earlier with a mixture of *dsRNase1*- and *-2*-specific dsRNAs, showing no or little digestion. (*b*) Box plot display of gel band fluorescence intensities of gut extracts mixed with dsRNA for 10 min from adults collected 24 h post dsRNA injection. (*c*) Box plot display of dsRNA band intensities following a 60 min exposure to gut extracts from adults collected 24 h post dsRNA injection. The means and standard deviations of eight replicate samples are presented. Different letters denote statistically different means (ANOVA with Tukey's *post hoc* test, $p < 0.05$).

stained with ethidium bromide, and dsRNA degradation was assessed by densitometry-based analyses. The *gus*-dsRNA was almost completely degraded within 10 min when incubated in gut extracts of adult *B. tryoni* (figure 2*a*). To evaluate the nuclease activity of *dsRNase1* and *dsRNase2*, young adults were injected into their haemocoel with dsRNA targeting *dsRNase1*, *dsRNase2* or 50 : 50 mixes of both nucleases. *DsRNase1* transcripts were reduced by 88.7% and 85.1% with single or mixed dsRNAs, respectively, 24 h post injection ($p < 0.05$ for both treatments), and by 48 h post injection they remained knocked down 84.0% ($p < 0.01$) and 71.0% ($p < 0.05$) (electronic supplementary material, figure S2). *DsRNase2* transcript levels were knocked down 92.8% ($p < 0.01$) and 86.9% ($p < 0.01$) in the single and mixed treatments, respectively, and knockdown remained strong after 48 h (97.6% with single dsRNA and 95.5% in the mixed treatments, $p < 0.02$).

The rate of *gus*-dsRNA degradation was also measured in dissected gut contents of adults injected with dsRNA either targeting one or both nucleases or targeting *gfp*. The *gus*-dsRNA was significantly protected against nuclease degradation in the first 10 min of exposure to gut extracts, for insects injected with dsRNA targeting *dsRNase1* or *dsRNase2* alone or in combination (figure 2*b*). The *gus*-dsRNA was still protected from complete nuclease degradation for up to 60 min when insects were injected with the mixture of dsRNAs targeting both *dsRNase1* and *dsRNase2* (figure 2*c*).

## 3.3. Knockdown of nuclease genes *in vivo*

We targeted a gene for which RNAi knockdown would not be lethal but would produce a detectable phenotype to test if knockdown of the two nucleases' transcripts could improve RNAi efficacy. In *D. melanogaster*, the *yellow* (*y*) gene encodes

royalsocietypublishing.org/journal/rsob Open Biol. 9: 190198

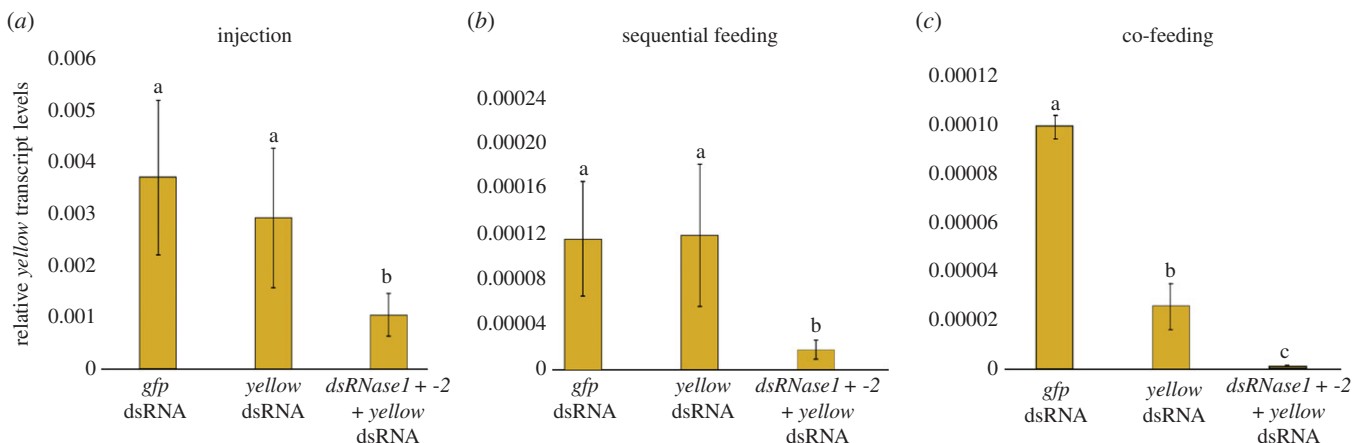

**Figure 3.** RNAi-mediated knockdown of *yellow* gene transcripts following different dsRNA delivery methods. (*a*) Adult females were injected with 1 µg of *gfp*- or a mixture of 0.5 µg each of *dsRNase1*- and *-2*-dsRNAs. The insects were then fed sugar water containing 2 µg of *gfp*- or *yellow*-dsRNA for 6 consecutive days, after which *yellow* transcript levels were assessed by quantitative RT-PCR. (*b*) The *yellow* transcripts were measured as described above, except adult insects were fed 2 µg of *gfp*- or 2 µg each of *dsRNase1*- and *-2*-dsRNAs for 3 days before being fed 2 µg of *yellow*-dsRNA for 3 days, or were fed 2 µg of *gfp*-dsRNA for 6 days as a control. (*c*) *Yellow* transcripts were assessed after adult insects were fed 2 µg of *gfp*- or *yellow*-dsRNA or were co-fed a mixture of 2 µg each of *yellow*-, *dsRNase1*- and *dsRNase2*-dsRNA dsRNAs for 6 consecutive days. Values represent the means and standard errors of eight biological replicates; different letters indicate significant differences (ANOVA with Tukey's *post hoc* test, $p < 0.05$).

an enzyme in the melanin biosynthesis pathway [35]. A search of the *B. tryoni* genome identified a putative *y* homologue with 49% nucleotide identity to that of the *Drosophila* gene, and 89% or higher identity to *yellow* genes in other *Bactrocera* species (electronic supplementary material, table S4). Newly eclosed adult flies were injected with a mixture of dsRNAs targeting both *dsRNase1* and *dsRNase2* or a control *gfp*-dsRNA, followed by oral delivery of *yellow*-dsRNA through daily doses of 2 µg of dsRNA in sugar water. After 6 days of dsRNA feeding, *dsRNase1* and *dsRNase2* transcript levels remained knocked down 85% ($p < 0.01$) and 88% ($p < 0.05$), respectively (electronic supplementary material, figure S3A), and *yellow* transcripts were reduced 82% ($p < 0.05$), relative to flies that had been injected with *gfp*-dsRNA (figure 3*a*). Adult flies were then sequentially fed dsRNAs, first with a mixture of the two nuclease-specific dsRNAs (or *gfp*-dsRNA as a negative control) for 3 days, followed by 3 days of feeding on *yellow*-dsRNA. At day 6, transcript levels of *dsRNase1* and *dsRNase2* were reduced 70% and 64% ($p < 0.05$; electronic supplementary material, figure S3B) and *yellow* transcripts were reduced 80% ($p < 0.001$), respectively, relative to *gfp*-dsRNA controls (figure 3*b*). Prior feeding with the non-specific *gfp*-dsRNA did not affect knockdown of *yellow* transcripts. Simultaneous oral delivery of dsRNAs targeting both nucleases and *yellow* genes for six consecutive days was also effective in knocking down transcripts of *yellow* 100% ($p < 0.0001$) (figure 3*c*). Knockdown of *dsRNase1* and *dsRNase2* in these treatments was 75% ($p < 0.01$) and 88% ($p < 0.05$), respectively, relative to *gfp*-dsRNA-treated controls (electronic supplementary material, figure S3C).

Persistence of RNAi-mediated knockdown from larvae to adults was evaluated through oral delivery of dsRNA by adding dsRNA to the larval diet daily until pupation. When larvae were fed *yellow*-dsRNA, knockdown of *yellow* was observed up to 10 days post eclosion relative to *gfp*-dsRNA controls, although the knockdown relative to the negative controls (fed *gfp*-dsRNA) diminished over time—80% knockdown at day 3, 68% at day 7 and 62% at day 10 (*t*-test, $p < 0.05$; figure 4). When dsRNA targeting *dsRNase1*

and *dsRNase2* was added to the food in addition to *yellow*-dsRNA, over 99% knockdown of *yellow* was achieved for up to at least 10 days post eclosion ($p < 0.01$). The nuclease genes' transcript levels were measured only on day 5 and were observed to be reduced 54.8% ($p < 0.05$) and 81.7% ($p < 0.01$) for *dsRNase1* and *dsRNase2*, respectively (electronic supplementary material, figure S4).

### 3.4. Effect of liposomes on dsRNA degradation and RNAi efficacy

Combining dsRNA with a liposome-based microcarrier significantly improved the extent of RNAi-mediated knockdown of targeted mRNAs in *ex vivo* gut extracts (electronic supplementary material, figure S5) and in adult flies. While adults feeding on naked *yellow*-dsRNA for 3 days showed a 50% reduction in transcripts relative to *gfp*-dsRNA controls ($p < 0.05$), adults fed *yellow*-dsRNAs encapsulated in liposomes for the same period showed a significantly greater knockdown of 95.2% ($p < 0.05$) (figure 5).

*Yellow* knockdown was further assessed through phenotypic assays. Since *yellow* in *Drosophila* encodes a protein within the melanin biosynthetic pathway [35], melanin production was used to assess the efficacy of RNAi treatments targeting the *yellow* gene's transcripts in *B. tryoni*. Haemolymph from flies that were fed naked *yellow* dsRNA for 6 days reduced melanin production by 59% over the 30 h period, relative to the *gfp*-dsRNA controls ($p < 0.05$), while haemolymph from flies fed *yellow*-dsRNA-liposome complexes showed no significant production of melanin over the same period, relative to control flies fed *gfp*-dsRNA-liposome complexes (figure 6).

## 4. Discussion

RNAi efficiencies can vary considerably in different insect species, and, in many instances, the lack of effective RNAi has been attributed to endogenous nucleases that can destroy the dsRNA before it can reach its intracellular targets

royalsocietypublishing.org/journal/rsob    Open Biol. 9: 190198

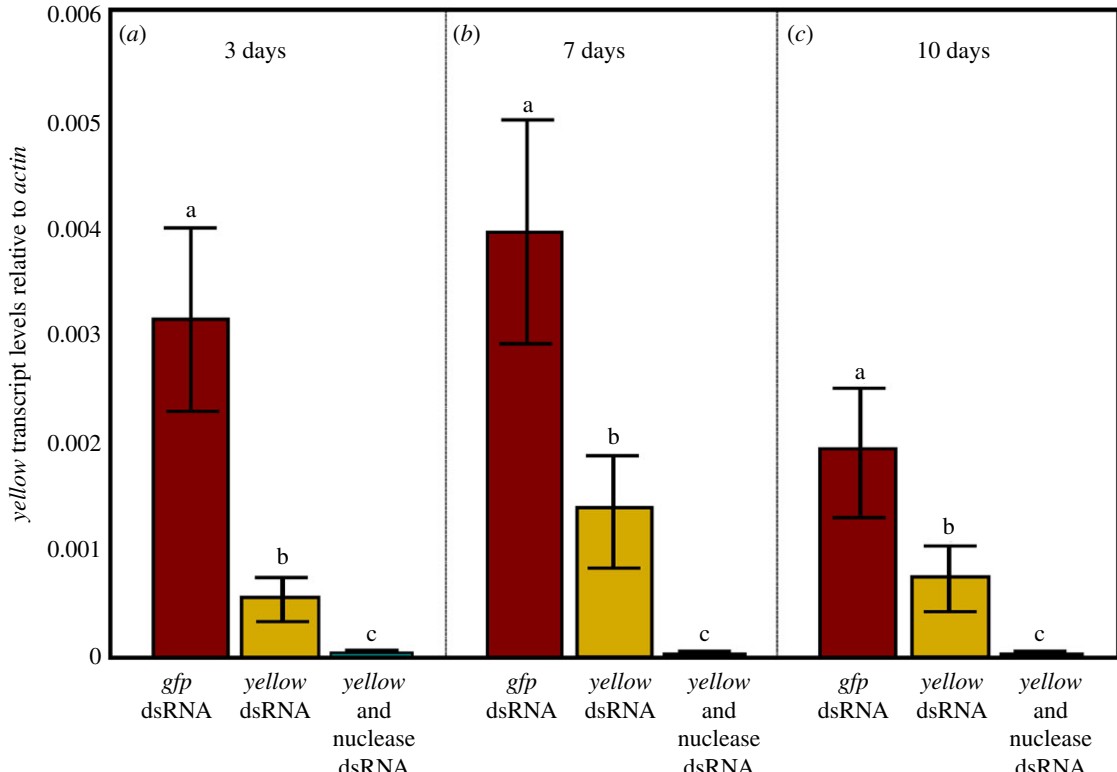

**Figure 4.** Persistence of knockdown of *yellow*-dsRNA into adulthood after oral delivery to larvae. Larvae were fed continuously on a diet containing 1 μg per larva of *gfp*-, *yellow*-, or *yellow* + nuclease-dsRNA for 4 days and 2 μg per larva for 3 days, for a total of 7 consecutive days of treatment. The resulting adults were assessed for *yellow* transcript levels at (*a*) day 3, (*b*) day 7 and (*c*) day 10 post eclosion. Values represent the means and standard errors of eight biological replicates; different letters indicate significant differences (ANOVA with Tukey's *post hoc* test, $p < 0.05$).

(reviewed in [15]). In this study, we observed that dsRNAs could be rapidly degraded by nucleases within the gut of the Queensland fruit fly, *B. tryoni*, but by protecting dsRNAs using liposome encapsulation or by knockdown of the nucleases through feeding the insects nuclease-specific dsRNAs it was possible to significantly improve RNAi efficacy in this insect.

Two nucleases were identified within the *B. tryoni* genome that were expressed either exclusively (*dsRNase1*) or predominantly (*dsRNase2*) within the gut of both larvae and adults. DsRNA exposed to the adult gut juices was quickly degraded, with almost 90% digested within the first 10 min. DsRNA degradation in the gut has been confirmed in a range of insects [13,14,36–38], and, in each case, the nuclease activity was observed to reduce the efficacy of RNAi. In our study, we observed that RNAi-mediated knockdown of either *dsRNase1* or *dsRNase2* provided moderate protection to the dsRNA, while knockdown of both dsRNases simultaneously provided almost complete protection of the dsRNA for the first 10 min of exposure to the gut extracts. Clearly, to achieve effective protection of dsRNAs with an insect's gut, it may be necessary to suppress the activity of multiple nucleases.

Protection of the dsRNA from gut nuclease activity was achieved using liposomes to encapsulate the dsRNAs. In the *ex vivo* treatments, liposome encapsulation protected greater than 95% of the dsRNA in the first 10 min of exposure and continued to provide protection to approximately 30% of the dsRNA for 60 min (electronic supplementary material, figure S5). Mixing liposome-encapsulated dsRNA in the larval diet resulted in visible aggregation of the liposome complexes in the food (perhaps due to the low pH of the diet), which the larvae avoided during feeding, and, hence,

a lack of RNAi-mediated knockdown. However, mixing liposome-encapsulated dsRNAs in the adult sucrose-based diet was simple and highly effective as a dsRNA delivery method. Adult insects feeding on liposome-encapsulated dsRNA showed considerably greater transcript knockdown of *yellow*, the target gene tested, than those feeding on non-encapsulated dsRNA. Various functions have been attributed to *yellow* proteins in insects [39], including melanin production, for both cuticle pigmentation and for haemocyte-mediated innate immune responses. Changes in pigmentation of dsRNA-treated adult insects were not observed, as their cuticles were already fully pigmented at the time of dsRNA feeding. However, the loss of melanization in the haemolymph of *yellow*-dsRNA-treated insects provided convincing evidence that (i) the *yellow* gene isolated here is associated with aspects of melanin biosynthesis and (ii) the liposome-mediated delivery of *yellow*-dsRNA greatly improved the RNAi efficacy. In the liposome-dsRNA-treated insects, the accumulation of melanin was effectively negligible after 30 h, which indicates that, at the phenotypic level, gene knockdown was virtually complete. These observations suggest that liposomes can indeed protect the dsRNA within the gut and improve delivery of intact dsRNA to the gut cells. The improved RNAi-mediated knockdown of *yellow*, which was assessed from haemolymph, indicates that the liposomes did not hinder the systemic spread of the dsRNA to cells beyond the gut. Similar improvements in RNAi efficacy using liposomes to protect the dsRNA have been observed in spotted wing *Drosophila* [40], cockroaches [41] and ticks [42], which suggests that this method of dsRNA delivery could improve RNAi efficiency in a broad range of invertebrates. Liposomes have the added advantage of protecting the dsRNA within an insect gut,

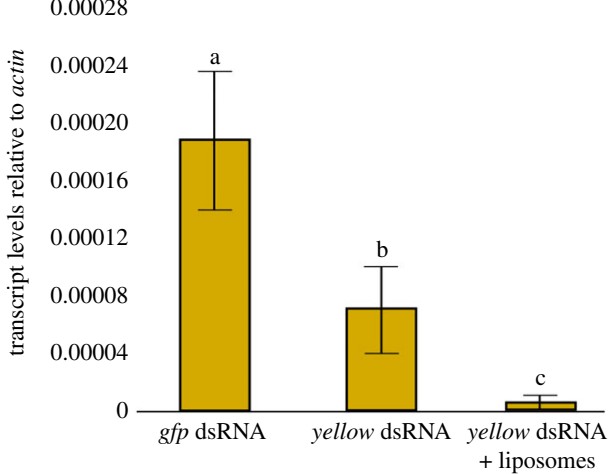

**Figure 5.** Knockdown of transcript levels in insects fed naked or liposome-encapsulated dsRNAs. Adult female flies were fed 2 μg of naked *gfp*- or *yellow*-dsRNA or were fed 2 μg of *yellow*-dsRNA mixed with liposomes for 3 days before RNA was extracted for quantitative RT-PCR analysis. The values represent the means and standard errors of eight biological replicates; different letters indicate significant differences, ANOVA with Tukey's *post hoc* test, $p < 0.05$.

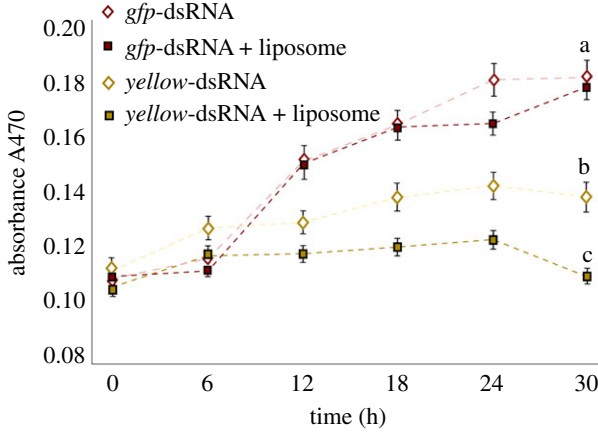

**Figure 6.** Reduced melanin production in adult females fed *yellow*-dsRNA. Flies were fed dsRNA or dsRNA in liposomes for 6 consecutive days before haemolymph was collected and assessed for melanin production. Pooled haemolymph from 10 flies was analysed using a BioTek Synergy H1 plate reader at 470 nm over a 30 h period. Negative control flies were fed 2 μg of naked or liposome-encapsulated *gfp*-dsRNA while experimental flies were fed 2 μg of naked or liposome-encapsulated *yellow*-dsRNA. The changes in absorbance over time are plotted, while the values at the 30 h time point were used to demonstrate a significant decrease in melanin production in the *yellow*-dsRNA-fed insects. The values represent the mean and standard errors for six replicates; different letters indicate significant differences, ANOVA with Tukey's *post hoc* test, $p < 0.05$.

without needing to identify the specific nucleases that could degrade the dsRNA. However, liposomes can be cytotoxic, depending on the target cell types [43], and, hence, multiple liposome formulations may need to be tested for efficacy in each species.

Commercially prepared liposomes are quite expensive, and until higher throughput liposome production methods are developed they would not be appropriate for large-scale dsRNA delivery applications in insects. A much more cost-effective method of protecting the dsRNA and improving RNAi efficacy was demonstrated with either sequentially or simultaneously feeding the insects nuclease-specific dsRNAs along with the *yellow*-dsRNA. In this study, we observed no significant knockdown of *yellow* transcripts when adults were fed either a non-specific dsRNA (*gfp*-dsRNA) or *yellow*-dsRNA alone after 3 days. While pre-exposure to non-specific dsRNAs has been observed to prime the RNAi machinery and thereby improve subsequent gene-specific RNAi-mediated knockdown in some insects [44], pre-treatment of *B. tryoni* with the non-specific *gfp*-dsRNA did not affect knockdown of *yellow* transcripts. In contrast, co-feeding the insects with the dsRNAs targeting both *dsRNase1* and *dsRNase2* along with the *yellow*-dsRNA resulted in almost complete (99%) knockdown of *yellow* transcripts. In larvae, knockdown of nucleases increased RNAi efficiency to almost 100%. Equally important, the transcript knockdown persisted for at least 10 days into adulthood, which could prove very useful for ensuring sustained RNAi phenotypes later in development.

A growing number of studies have demonstrated that nucleases are an important barrier to successful application of RNAi in some insects. Here, we demonstrated that either liposome encapsulation of dsRNA or co-feeding nuclease-specific dsRNAs with target dsRNAs can greatly improve the efficacy of RNAi in *B. tryoni*. For those species where gut nucleases are impairing the uptake of dsRNA, minimizing nuclease degradation of the dsRNA could be an effective solution to what is likely to be a common problem with RNAi applications in insects. If RNAi-mediated reduction of nuclease activity is to be used more widely in

other insects for improved RNAi efficiency, it will require a clear understanding of which nucleases are found within the gut of each target species. Where this improved RNAi efficacy may be realized is in large-scale applications such as SIT and foliar insecticides. Relevant targets for these applications include genes regulating male fertility for SIT or essential genes for insecticidal applications. Achieving near 100% efficacy of both methods is highly desirable; sub-lethal doses of insecticidal dsRNAs, for example, could hasten the development of resistance, while incomplete male sterility could reduce the efficiency of SIT-based control. Previously, we demonstrated that RNAi-mediated knockdown of spermatogenesis genes reduced male fecundity by 78% in *B. tryoni* [10], but for many tephritid SIT programmes radiation-induced sterility rates of greater than 99% are the norm [45]. In future studies, we intend to examine whether co-delivery of nuclease- and testis-specific dsRNAs can increase the male sterility rate closer to field application standards, as an alternative to radiation-based sterilization in SIT-mediated control strategies. Knockdown of nuclease activity may not only improve RNAi efficacy in insect control applications, but could prove helpful in increasing knockdown of transcripts to validate gene function in basic molecular biology studies.

**Data accessibility.** All data not provided in the main text are provided in the electronic supplementary material. (We are willing to share all raw data from all figures within the paper if needed, but we are uncertain what additional data would be required.)

**Authors' contributions.** S.W. and A.T. conceptualized and designed the study, analysed all data and wrote the manuscript. A.T., J.Y.P., D.H. and D.G. conducted the experiments, analysed and interpreted data and critically reviewed the manuscript. S.W. funded the project and provided coordination and supervision throughout.

**Competing interests.** We declare we have no competing interests.

**Funding.** The project Dietary Sterilization of Male Queensland Fruit Fly (grant no. AI13001) is funded by Hort Innovation.

**Acknowledgements.** Thanks to Dr Solomon Balagawi (Elizabeth Macarthur Agriculture Institute, Australia) for providing the insects used in this study. The authors are grateful to Chris Hardy and John Oakeshott (CSIRO Australia) for invaluable commentary on the manuscript. This research was conducted as part of the SITplus collaborative fruit fly program.

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
