## [Reviewer comments · Open Biology]

Review History

RSOB-19-0198.R0 (Original submission)

Review form: Reviewer 1

Recommendation

Major revision is needed (please make suggestions in comments)

Do you have any ethical concerns with this paper?

No

Comments to the Author

In this study, Tayler et al. investigate the involvement of dsRNases in the observed reduced RNAi efficacy in the tephritid fruit fly *Bactrocera tryoni*. First, a bioinformatics analysis is performed to identify different potential dsRNases in the genome of this pest insect. Next, the expression of these dsRNases is evaluated, confirming expression in the gut. An ex vivo dsRNA stability assay is then conducted in gut extract and finally, in a series of so-called RNAi-of-RNAi in vivo experiments the dsRNases are knocked down to evaluate whether this improves RNAi efficacy.

I think the experimental design in this study is for the mostpart appropriate and containing the proper controls. The data also support the conclusions made by the authors in my opinion. However, I do have two issues with the experimental design. One is the use of only one reference gene in the qPCR experiments and the other relates to an additional control which could have been included in the RNAi-of-RNAi experiments (see next two comments for more detailed info)

All qPCR experiments in this study have been conducted using only one reference gene. Both for expression profiling and RNAi experiments, it is recommended these days to use at least 2 and preferably even 3 or more stable reference genes for qPCR analysis. As is also indicated in the MIQE guidelines. Reference genes stability can be very variable between different experimental conditions, and this can lead to a significant bias in the results. In this case, you are evaluating expression data between different tissues, between different life stages and between different experimental treatments (in the case of the RNAi experiments). All of these can have an influence on the expression of the reference gene itself. Especially for expression profiling this is very important. I would like to see some support that the reference gene (actin) was stable between the different experimental conditions and ideally, I would like to see the addition of at least one more stable reference.

In the RNAi-of-RNAi studies, I think it would have been interesting to add a treatment whereby the insects were first injected or fed with a non-specific dsRNA before being fed with the yellow dsRNA. This as a control for the RNAi machinery stimulation effect non-specific dsRNA can have. Certain studies have shown that injection of non-specific dsRNA (for example dsGFP) can already increase efficiency of a follow-up RNAi experiment with specific dsRNA. However, I do not consider the control to be vital in this case since all data taken together do point towards a specific nuclease-dependent effect, rather than a non-specific effect. Nonetheless, it could have an effect as well and could therefore perhaps just be addressed in the discussion.

In the last paragraph of the Introduction, the authors already present quite a lot of the results. Personally I am not in favor of already describing results here. I prefer to just have the aim of the study (and chosen strategy) described here, without results. But I will leave it up to the authors to decide on that. Just a personal preference.

The authors write that the dsRNAs for the dsRNases, GFP and *gus* were purchased commercially. Do you know whether the dsRNA was produced microbially? What do you know about the purity of this dsRNA? I think it would be useful information to add.

In most studies where *ex vivo* experiments are done in gut juice, the gut contents are being extracted/collected by centrifugation. I was wondering whether there was any particular reason why this is not the case here. Also, the authors mention that three different methods were tested; using intact gut, using gut cut in pieces and using homogenized gut. I was a bit surprised to read that there was no difference in degradation pattern between these three. I assume homogenization should also release most intracellular enzymes? Including Dicer-2 for example. You would think this has an effect on the (speed of) degradation. In any case, I am happy that the authors eventually chose for the gut tissue which was just cut in pieces.

Regarding the phylogenetic tree, I was wondering why the authors decided to produce two separate phylo trees for both dsRNases. Wouldn't it have been more interesting to combine this in one tree? Especially since some other insect species contain more than two of these dsRNases. It would be stronger evidence I think of the homology of these RNases within the insect clade. Also, there are a number of other nucleases that could be added here. Or are these nucleases really too divergent to be analysed in the same tree? Or perhaps there is another reason to separate them?

I was also wondering to what extent differences between experiments could be related to differences in doses to which the flies were exposed to. Differences between experiments, differences between larvae and adults, between injection and feeding, etc. It's not always easy to follow the doses which the flies have taken up (either orally or by injection) between the different experiments. Perhaps it would not be a bad idea to have these included in the figure legends? Or perhaps have some sort of overview table included in the Material and Methods with the doses and durations? Or it could be included in the Results section with some of the results in the table perhaps. Just a suggestion.

Some minor comments:

- First mention in the abstract, the species is written 'tyroni' instead of tryoni
- Be consistent with the spelling of dsRNases (they are written as dsRNAse in some of the figures for example)
- There is a recent publication conducting a similar study on another coleopteran species (Prentice et al., 2019). This one could perhaps be included in the intro or discussion as well.
- Line 360: equally important

Review form: Reviewer 2

Recommendation

Accept with minor revision (please list in comments)

Do you have any ethical concerns with this paper?

No

Comments to the Author

The manuscript by Tayler et al. demonstrates two technical advances to improve the knockdown efficiency of exogenously added double-stranded (ds) RNAs in the fruit fly agricultural pest, *Bactrocera tryoni*.

Specifically, the authors show that *B. tryoni* gut expresses two related dsRNA degrading enzymes, "dsRNAases". They show that knockdown of these dsRNAases (via dsRNA against these dsRNAases!) increases the knockdown efficiency of ingested dsRNA against yellow, a melanization gene that can be measured by RTqPCR in the fly hemolymph. In addition, the authors show that feeding *B. tryoni* liposomes with dsRNA also increases the knockdown efficiency of the dsRNA.

The data appear convincing and the use of multiple biological replicates is appreciated. However, the biological significance of the advances is unclear. For example, the authors showed that while the efficiency of RNAi in *B. tryoni* can be improved, it is unclear if it needs to be, since it was already effective at knocking down the RNAases. A more convincing readout would be a biologically relevant target, such as male sterility.

Lastly, there are some typos in the article that should be fixed. For example, in the sentence on lines 289-290 -- a number is missing. Also, the legend for Figure 3 is tricky to read: dsRNA against yellow was added twice? It is unclear if it is a typo. If not, a minor rewrite would be helpful.

Decision letter (RSOB-19-0198.R0)

24-Sep-2019

Dear Dr Whyard:

We are writing to inform you that the Editor has reached a decision on your manuscript RSOB-19-0198 entitled "Efficiency of RNA interference is improved by knockdown of dsRNA nucleases in tephritid fruit flies.", submitted to Open Biology.

As you will see from the reviewers' comments below, there are a number of criticisms that prevent us from accepting your manuscript at this stage. The reviewers suggest, however, that a revised version could be acceptable, if you are able to address their concerns. If you think that you can deal satisfactorily with the reviewer's suggestions, we would be pleased to consider a revised manuscript.

The revision will be re-reviewed, where possible, by the original referees. As such, please submit the revised version of your manuscript within four weeks. If you do not think you will be able to meet this date please let us know immediately.

When submitting your revised manuscript, please respond to the comments made by the referee(s) and upload a file "Response to Referees" in "Section 6 - File Upload". You can use this to document any changes you make to the original manuscript. In order to expedite the processing of the revised manuscript, please be as specific as possible in your response to the referee(s).

Please see our detailed instructions for revision requirements
<https://royalsociety.org/journals/authors/author-guidelines/>

Sincerely,
The Open Biology Team
mailto: openbiology@royalsociety.org

Editor's Comments to Author(s):

Reviewer(s)' Comments to Author(s):
Referee: 1

Comments to the Author(s)

In this study, Tayler et al. investigate the involvement of dsRNases in the observed reduced RNAi efficacy in the tephritid fruit fly *Bactrocera tryoni*. First, a bioinformatics analysis is

performed to identify different potential dsRNases in the genome of this pest insect. Next, the expression of these dsRNases is evaluated, confirming expression in the gut. An *ex vivo* dsRNA stability assay is then conducted in gut extract and finally, in a series of so-called RNAi-of-RNAi *in vivo* experiments the dsRNases are knocked down to evaluate whether this improves RNAi efficacy.

I think the experimental design in this study is for the mostpart appropriate and containing the proper controls. The data also support the conclusions made by the authors in my opinion. However, I do have two issues with the experimental design. One is the use of only one reference gene in the qPCR experiments and the other relates to an additional control which could have been included in the RNAi-of-RNAi experiments (see next two comments for more detailed info)

All qPCR experiments in this study have been conducted using only one reference gene. Both for expression profiling and RNAi experiments, it is recommended these days to use at least 2 and preferably even 3 or more stable reference genes for qPCR analysis. As is also indicated in the MIQE guidelines. Reference genes stability can be very variable between different experimental conditions, and this can lead to a significant bias in the results. In this case, you are evaluating expression data between different tissues, between different life stages and between different experimental treatments (in the case of the RNAi experiments). All of these can have an influence on the expression of the reference gene itself. Especially for expression profiling this is very important. I would like to see some support that the reference gene (actin) was stable between the different experimental conditions and ideally, I would like to see the addition of at least one more stable reference.

In the RNAi-of-RNAi studies, I think it would have been interesting to add a treatment whereby the insects were first injected or fed with a non-specific dsRNA before being fed with the yellow dsRNA. This as a control for the RNAi machinery stimulation effect non-specific dsRNA can have. Certain studies have shown that injection of non-specific dsRNA (for example dsGFP) can already increase efficiency of a follow-up RNAi experiment with specific dsRNA. However, I do not consider the control to be vital in this case since all data taken together do point towards a specific nuclease-dependent effect, rather than a non-specific effect. Nonetheless, it could have an effect as well and could therefore perhaps just be addressed in the discussion.

In the last paragraph of the Introduction, the authors already present quite a lot of the results. Personally I am not in favor of already describing results here. I prefer to just have the aim of the study (and chosen strategy) described here, without results. But I will leave it up to the authors to decide on that. Just a personal preference.

The authors write that the dsRNAs for the dsRNases, GFP and gus were purchased commercially. Do you know whether the dsRNA was produced microbially? What do you know about the purity of this dsRNA? I think it would be useful information to add.

In most studies where *ex vivo* experiments are done in gut juice, the gut contents are being extracted/collected by centrifugation. I was wondering whether there was any particular reason why this is not the case here. Also, the authors mention that three different methods were tested; using intact gut, using gut cut in pieces and using homogenized gut. I was a bit surprised to read that there was no difference in degradation pattern between these three. I assume homogenization should also release most intracellular enzymes? Including Dicer-2 for example. You would think this has an effect on the (speed of) degradation. In any case, I am happy that the authors eventually chose for the gut tissue which was just cut in pieces.

Regarding the phylogenetic tree, I was wondering why the authors decided to produce two separate phylo trees for both dsRNases. Wouldn't it have been more interesting to combine this

in one tree? Especially since some other insect species contain more than two of these dsRNases. It would be stronger evidence I think of the homology of these RNases within the insect clade. Also, there are a number of other nucleases that could be added here. Or are these nucleases really too divergent to be analysed in the same tree? Or perhaps there is another reason to separate them?

I was also wondering to what extent differences between experiments could be related to differences in doses to which the flies were exposed to. Differences between experiments, differences between larvae and adults, between injection and feeding, etc. It's not always easy to follow the doses which the flies have taken up (either orally or by injection) between the different experiments. Perhaps it would not be a bad idea to have these included in the figure legends? Or perhaps have some sort of overview table included in the Material and Methods with the doses and durations? Or it could be included in the Results section with some of the results in the table perhaps. Just a suggestion.

Some minor comments:

- First mention in the abstract, the species is written 'tyroni' instead of tryoni
- Be consistent with the spelling of dsRNases (they are written as dsRNase in some of the figures for example)
- There is a recent publication conducting a similar study on another coleopteran species (Prentice et al., 2019). This one could perhaps be included in the intro or discussion as well.
- Line 360: equally important

Referee: 2

Comments to the Author(s)

The manuscript by Tayler et al. demonstrates two technical advances to improve the knockdown efficiency of exogenously added double-stranded (ds) RNAs in the fruit fly agricultural pest, *Bactrocera tryoni*.

Specifically, the authors show that *B. tryoni* gut expresses two related dsRNA degrading enzymes, "dsRNAases". They show that knockdown of these dsRNAases (via dsRNA against these dsRNAases!) increases the knockdown efficiency of ingested dsRNA against yellow, a melanization gene that can be measured by RTqPCR in the fly hemolymph. In addition, the authors show that feeding *B. tryoni* liposomes with dsRNA also increases the knockdown efficiency of the dsRNA.

The data appear convincing and the use of multiple biological replicates is appreciated. However, the biological significance of the advances is unclear. For example, the authors showed that while the efficiency of RNAi in *B. tryoni* can be improved, it is unclear if it needs to be, since it was already effective at knocking down the RNAases. A more convincing readout would be a biologically relevant target, such as male sterility.

Lastly, there are some typos in the article that should be fixed. For example, in the sentence on lines 289-290 -- a number is missing. Also, the legend for Figure 3 is tricky to read: dsRNA against yellow was added twice? It is unclear if it is a typo. If not, a minor rewrite would be helpful.

Author's Response to Decision Letter for (RSOB-19-0198.R0)

See Appendix A.

RSOB-19-0198.R1 (Revision)

Review form: Reviewer 1

Recommendation

Accept as is

Do you have any ethical concerns with this paper?

No

Comments to the Author

I am happy with the answers provided by the authors to my questions and comments and with the changes made to the manuscript.

Review form: Reviewer 2

Recommendation

Accept with minor revision (please list in comments)

Do you have any ethical concerns with this paper?

No

Comments to the Author

The authors state "We have addressed all of the questions and concerns of the two referees." It is unclear to this reviewer that the authors addressed all the concerns raised. A point-by-point response would have been appreciated.

Decision letter (RSOB-19-0198.R1)

25-Oct-2019

Dear Dr Whyard

We are pleased to inform you that your manuscript RSOB-19-0198.R1 entitled "Efficiency of RNA interference is improved by knockdown of dsRNA nucleases in tephritid fruit flies." has been accepted by the Editor for publication in Open Biology. The reviewer(s) have recommended publication, but also ask for some clarification. Therefore, we invite you to respond to the reviewer 2 comments. In particular, there was a "point-by-point" response to the original reviewer comments missing in your revised submission. Can you please provide this document to allow the reviewer and the editorial staff to directly assess your specific responses to the original reviewer comments.

Please submit the revised version of your manuscript within 14 days. If you do not think you will be able to meet this date please let us know immediately and we can extend this deadline for you.

- 1) A text file of the manuscript (doc, txt, rtf or tex), including the references, tables (including captions) and figure captions. Please remove any tracked changes from the text before submission. PDF files are not an accepted format for the "Main Document".
- 2) A separate electronic file of each figure (tiff, EPS or print-quality PDF preferred). The format should be produced directly from original creation package, or original software format. Please note that PowerPoint files are not accepted.
- 3) Electronic supplementary material: this should be contained in a separate file from the main text and meet our ESM criteria (see <http://royalsocietypublishing.org/instructions-authors#question5>). All supplementary materials accompanying an accepted article will be treated as in their final form. They will be published alongside the paper on the journal website and posted on the online figshare repository. Files on figshare will be made available approximately one week before the accompanying article so that the supplementary material can be attributed a unique DOI.

Online supplementary material will also carry the title and description provided during submission, so please ensure these are accurate and informative. Note that the Royal Society will not edit or typeset supplementary material and it will be hosted as provided. Please ensure that the supplementary material includes the paper details (authors, title, journal name, article DOI). Your article DOI will be 10.1098/rsob.2016[last 4 digits of e.g. 10.1098/rsob.20160049].

- 4) A media summary: a short non-technical summary (up to 100 words) of the key findings/importance of your manuscript. Please try to write in simple English, avoid jargon, explain the importance of the topic, outline the main implications and describe why this topic is newsworthy.

Images

Data-Sharing

It is a condition of publication that data supporting your paper are made available. Data should be made available either in the electronic supplementary material or through an appropriate repository. Details of how to access data should be included in your paper. Please see <http://royalsocietypublishing.org/site/authors/policy.xhtml#question6> for more details.

Data accessibility section

Sincerely,

The Open Biology Team

<mailto:openbiology@royalsociety.org>

Reviewer(s)' Comments to Author:

Referee: 1

Comments to the Author(s)

I am happy with the answers provided by the authors to my questions and comments and with the changes made to the manuscript.

Referee: 2

Comments to the Author(s)

The authors state "We have addressed all of the questions and concerns of the two referees." It is unclear to this reviewer that the authors addressed all the concerns raised. A point-by-point response would have been appreciated.

"I am surprised by the authors' decision to not respond in a point-by-point fashion to the reviewer comments. In looking over their revised manuscript, it appears that they did not respond to the questions that I raised about whether improving RNAi efficiency was needed, since it was RNAi that led to the improvement of RNAi in the first place. Similarly, they did little to respond to concerns to reviewer #1 concerning controls. The revised manuscript does not increase my enthusiasm for the paper.

I think the outstanding question is: is this article of sufficient interest. If the authors had adjusted their argument to show they were solving a biologically relevant problem, I would think it would be of sufficient interest. However, it is unclear whether the increase in RNAi efficiency is necessary since RNAi was effective at increasing RNAi. Conversely, it is unclear if the new level of RNAi efficiency would be sufficient to achieve the goals of the researchers/field, e.g., to produce male sterility." Please address these two paragraphs as well in the point by point response. Thanks.

Author's Response to Decision Letter for (RSOB-19-0198.R1)

See Appendix B.

Decision letter (RSOB-19-0198.R2)

30-Oct-2019

Dear Mrs Tayler

We are pleased to inform you that your manuscript entitled "Efficiency of RNA interference is improved by knockdown of dsRNA nucleases in tephritid fruit flies." has been accepted by the Editor for publication in Open Biology.

Article processing charge

Please note that the article processing charge is immediately payable. A separate email will be sent out shortly to confirm the charge due. The preferred payment method is by credit card; however, other payment options are available.

Sincerely,

The Open Biology Team
mailto: openbiology@royalsociety.org

Appendix A

Response to Referees:

We would like to thank the reviewers for their insightful comments on the manuscript. Below is our response to the issues raised in the review (printed in italics).

Referee 1:

*In this study, Tayler et al. investigate the involvement of dsRNases in the observed reduced RNAi efficacy in the tephritid fruit fly *Bactrocera tryoni*. First, a bioinformatics analysis is performed to identify different potential dsRNases in the genome of this pest insect. Next, the expression of these dsRNases is evaluated, confirming expression in the gut. An ex vivo dsRNA stability assay is then conducted in gut extract and finally, in a series of so-called RNAi-of-RNAi in vivo experiments the dsRNases are knocked down to evaluate whether this improves RNAi efficacy.*

Response: We thank the reviewer for this evaluation.

I think the experimental design in this study is for the mostpart appropriate and containing the proper controls. The data also support the conclusions made by the authors in my opinion. However, I do have two issues with the experimental design. One is the use of only one reference gene in the qPCR experiments and the other relates to an additional control which could have been included in the RNAi-of-RNAi experiments (see next two comments for more detailed info)

*All qPCR experiments in this study have been conducted using only one reference gene. Both for expression profiling and RNAi experiments, it is recommended these days to use at least 2 and preferably even 3 or more stable reference genes for qPCR analysis. As is also indicated in the MIQE guidelines. Reference genes stability can be very variable between different experimental conditions, and this can lead to a significant bias in the results. In this case, you are evaluating expression data between different tissues, between different life stages and between different experimental treatments (in the case of the RNAi experiments). All of these can have an influence on the expression of the reference gene itself. Especially for expression profiling this is very important. I would like to see some support that the reference gene (*actin*) was stable between the different experimental conditions and ideally, I would like to see the addition of at least one more stable reference.*

Response: We thank the reviewer for noting the increased use of multiple reference genes when analyzing qRT-PCR data. We chose to use the *actin* reference gene because it is expressed in similar levels among tissues and developmental stages, and has been used as a reliable reference in multiple insect and Dipteran species (Lü et al. 2018). Further, it has been evaluated for stable expression in closely related *Bactrocera* species using various analysis software packages (Lü et al. 2018), and was determined to be the most reliable for gut tissue in *Bactrocera dorsalis* (Shen et al. 2010). Additionally, we analyzed primer efficiencies of the *actin* gene among the various tissues and developmental stages and determined they were within 1.5% of one another (96.0%-97.5%), and within 2.5% of the other genes within a given tissue. With such similar efficiencies, a single reference gene was considered appropriate for our comparisons of gene knockdown. Furthermore, the measurements of reduced

nuclease activity (due to treatments of nuclease-specific dsRNA) and reduced melanisation (due to treatments with yellow-specific dsRNA) provide further support that RNAi-mediated knockdown had occurred. Altogether, the results provide evidence that nucleases play a strong role in dsRNA degradation and RNAi efficacy. We have noted that multiple other papers recently published in this journal have used a single reference gene for qRT-PCR normalization in various organisms and tissues, including one which examined multiple insect tissues and developmental stages (<https://doi.org/10.1098/rsob.180158>). We have edited the methods to include these details and cited the three studies (Lü et al., 2018; Shen et al., 2010; Prentice et al., 2019). For all these reasons, we believe that the one reference gene was appropriate and sufficient for the analyses.

In the RNAi-of-RNAi studies, I think it would have been interesting to add a treatment whereby the insects were first injected or fed with a non-specific dsRNA before being fed with the yellow dsRNA. This as a control for the RNAi machinery stimulation effect non-specific dsRNA can have. Certain studies have shown that injection of non-specific dsRNA (for example dsGFP) can already increase efficiency of a follow-up RNAi experiment with specific dsRNA. However, I do not consider the control to be vital in this case since all data taken together do point towards a specific nuclease-dependent effect, rather than a non-specific effect. Nonetheless, it could have an effect as well and could therefore perhaps just be addressed in the discussion.

Response: That is a good point about potential non-specific effects of dsRNA on the RNAi machinery. While we did not **inject** adults with *gfp* dsRNA prior to feeding them *yellow* dsRNA, we performed a sequential **feeding** assay, where adults were fed *gfp* for 3 consecutive days followed by 3 days of *yellow* dsRNA and did not observe any increased/decreased efficiency of RNAi impacts on the *yellow* gene (Figure 3). We have added a comment in the Results section highlighting this point. We have also edited the methods to clarify the doses applied in all treatments for the sequential feeding assays.

In the last paragraph of the Introduction, the authors already present quite a lot of the results. Personally I am not in favor of already describing results here. I prefer to just have the aim of the study (and chosen strategy) described here, without results. But I will leave it up to the authors to decide on that. Just a personal preference.

Response: We agree that for some papers, it is nicer to leave out any summary of the research findings until the Discussion. However, in this paper, for which there are quite a few Figures provided, we felt that it would be helpful to provide an overview of the main findings. This prelude was meant to prepare the readers more fully for their evaluation of all the data.

The authors write that the dsRNAs for the dsRNases, GFP and gus were purchased commercially. Do you know whether the dsRNA was produced microbially? What do you know about the purity of this dsRNA? I think it would be useful information to add.

Response: The dsRNA purchased from AgroRNA uses a proprietary dsRNA synthesis platform and offers 2 grades of purity. We purchased the grade with the highest purity available, which provided standard

desalting procedures. We have included a comment in the Materials and Methods section.

In most studies where ex vivo experiments are done in gut juice, the gut contents are being extracted/collected by centrifugation. I was wondering whether there was any particular reason why this is not the case here. Also, the authors mention that three different methods were tested; using intact gut, using gut cut in pieces and using homogenized gut. I was a bit surprised to read that there was no difference in degradation pattern between these three. I assume homogenization should also release most intracellular enzymes? Including Dicer-2 for example. You would think this has an effect on the (speed of) degradation. In any case, I am happy that the authors eventually chose for the gut tissue which was just cut in pieces.

Response: We appreciate the observation about collecting gut contents by centrifugation and recognize that while centrifugation was performed with our samples, we neglected to include this step in our methods. The methods have been revised to include this information. We agree that homogenization of the gut should indeed release intracellular enzymes such as Dicer (which would inherently degrade the dsRNA), while in theory, intact guts should not release intracellular nucleases. Based on our knockdown results of the two dsRNases, these two enzymes appear to play the primary role in digestion of nucleases within the gut, at least while the dsRNA resides within the lumen. We have added a point of clarification in the Results section addressing this issue.

Regarding the phylogenetic tree, I was wondering why the authors decided to produce two separate phylo trees for both dsRNases. Wouldn't it have been more interesting to combine this in one tree? Especially since some other insect species contain more than two of these dsRNases. It would be stronger evidence I think of the homology of these RNases within the insect clade. Also, there are a number of other nucleases that could be added here. Or are these nucleases really too divergent to be analysed in the same tree? Or perhaps there is another reason to separate them?

Response: We did in fact try to merge the two nuclease gene trees, with the hope of simplifying the presentation of the data, but the low similarity (51%) between *B. tryoni* nuclease 1 and nuclease 2 resulted in less informative phylogenetic trees (with low bootstrap values). The two nucleases do not share any highly conserved regions with each other, but they do share highly conserved regions with homologues in other insects. By constructing two phylogenetic trees, we are able to align trimmed sequences of these conserved regions to illustrate their relationship with other nucleases more clearly. We have added a comment in the results section highlighting the differences between the two nucleases.

I was also wondering to what extent differences between experiments could be related to differences in doses to which the flies were exposed to. Differences between experiments, differences between larvae and adults, between injection and feeding, etc. It's not always easy to follow the doses which the flies have taken up (either orally or by injection) between the different experiments. Perhaps it would not be a bad idea to have these included in the figure legends? Or perhaps have some sort of overview table included in the Material and Methods with the doses and durations? Or it could be included in the Results section with some of the results in the table perhaps. Just a suggestion.

Response: Thank you for bringing it to our attention that the doses used in each treatment were not clear. We have edited the Methods and Figure captions for clarification.

Some minor comments:

- First mention in the abstract, the species is written 'tyroni' instead of tryoni

Response: Corrected

- Be consistent with the spelling of dsRNases (they are written as dsRNAse in some of the figures for example)

Response: Corrected

- There is a recent publication conducting a similar study on another coleopteran species (Prentice et al., 2019). This one could perhaps be included in the intro or discussion as well.

Response: Thank you for bringing that paper to our attention. We have added it to the Introduction and to the Methods section.

- Line 360: equally important

Response: Corrected

Referee 2:

General comments:

*The manuscript by Tayler et al. demonstrates two technical advances to improve the knockdown efficiency of exogenously added double-stranded (ds) RNAs in the fruit fly agricultural pest, *Bactrocera tryoni*.*

*Specifically, the authors show that *B. tryoni* gut expresses two related dsRNA degrading enzymes, "dsRNAases". They show that knockdown of these dsRNAases (via dsRNA against these dsRNAases!) increases the knockdown efficiency of ingested dsRNA against yellow, a melanization gene that can be measured by RTqPCR in the fly hemolymph. In addition, the authors show that feeding *B. tryoni* liposomes with dsRNA also increases the knockdown efficiency of the dsRNA.*

Response: We would like to thank the reviewer for this evaluation.

*The data appear convincing and the use of multiple biological replicates is appreciated. However, the biological significance of the advances is unclear. For example, the authors showed that while the efficiency of RNAi in *B. tryoni* can be improved, it is unclear if it needs to be, since it was already effective at knocking down the RNAases. A more convincing readout would be a biologically relevant target, such as male sterility.*

Response: We agree that making use of this improvement to RNAi efficacy has potential for a broad range of applications using other, perhaps more biologically relevant, gene targets, and in fact, we are currently conducting experiments examining the utility of dual knockdown of male fertility and nuclease genes. These studies will be the focus of another manuscript.

Lastly, there are some typos in the article that should be fixed. For example, in the sentence on lines 289-290 -- a number is missing. Also, the legend for Figure 3 is tricky to read: dsRNA against yellow was added twice? It is unclear if it is a typo. If not, a minor rewrite would be helpful.

Response: Corrected.

Appendix B

Response to Reviewers:

We thank the reviewers for the additional time spent reviewing our manuscript and appreciate their thoughtful comments and efforts towards improving our manuscript. We would like to offer our sincerest apologies to referee 2 who was concerned that not all of their questions were appropriately addressed in the first round of revisions, and regret that this second round of revisions will take a little more of their valuable time. We had not meant to disregard their suggestions in the first round; we were only thinking about how to keep the Discussion streamlined. We have given all recommendations a much more careful review and have endeavoured to address the remaining concerns in this latest revision. Please find all questions and from the second round listed below, now clearly numbered (as requested), for easier review.

Referee: 1

Comments to the Author(s)

1) I am happy with the answers provided by the authors to my questions and comments and with the changes made to the manuscript.

Response 1: We thank the referee for their time and effort reviewing this manuscript.

Referee: 2

Comments to the Author(s)

1) The authors state "We have addressed all of the questions and concerns of the two referees." It is unclear to this reviewer that the authors addressed all the concerns raised. A point-by-point response would have been appreciated.

Response 1: We sincerely apologize that not all concerns were addressed to your satisfaction. It is our hope that the edits provided in this revision are acceptable. While we tried to respond to first round questions in a point-by-point manner, we regret that we missed some key points in the previous round. We have numbered each point in this response and have included manuscript line numbers of the revisions performed in the current and previous responses to reviewers.

2) "I am surprised by the authors' decision to not respond in a point-by-point fashion to the reviewer comments. In looking over their revised manuscript, it appears that they did not respond to the questions that I raised about whether improving RNAi efficiency was needed, since it was RNAi that led to the improvement of RNAi in the first place."

Response 2: We again apologize that this was not directly addressed in the manuscript. We had interpreted the comments (point 2 of referee 2 in the previous response to referees) to imply that the choice of the target gene did not adequately address our primary objective, which was

to demonstrate that RNAi efficacy could be improved by knockdown of nucleases. We indicated in the previous response that other targets, including ones associated with male fertility, were the focus of some of our new studies, and hence, we did not have those data ready to present here.

We recognize that we missed the reviewer's main point, which was to provide evidence that improved RNAi efficacy is worth achieving in this species. We have now revised the manuscript to address the concern about why improving RNAi efficiency was needed (as there is definitely a lack of RNAi efficacy in adults fed only yellow dsRNA, but significant knockdown occurred when nuclease dsRNAs were provided (lines 368-377; 385-397; Figure 3)). Additionally, targeting more biologically relevant genes, such as those associated with male fertility for SIT applications, would demand a better efficacy than what we had demonstrated in one of our previous publications (now addressed in lines 385-397).

3) Similarly, they did little to respond to concerns to reviewer #1 concerning controls. The revised manuscript does not increase my enthusiasm for the paper.

Response 3: While we did not originally include clarification in the discussion, we did revise the methods to state that prior feeding with a non-specific dsRNA had no significant impact on the knockdown of *yellow* transcripts (lines 292-293). In this latest revision, we have now provided further clarification in the discussion (lines 368-374), where we explain that when insects were administered a non-specific dsRNA as a control, we observed no evidence of priming of the RNAi machinery that could enhance yellow knockdown; only administration of the nuclease-specific dsRNAs improved the efficacy of RNAi.

4) I think the outstanding question is: is this article of sufficient interest. If the authors had adjusted their argument to show they were solving a biologically relevant problem, I would think it would be of sufficient interest.

Response 4: We have revised our discussion to highlight the requirement of improved RNAi efficacy for various applications, and proposed biologically relevant targets and applications (with emphasis on SIT) that would stand to benefit from this improvement (lines 385-397).

5) However, it is unclear whether the increase in RNAi efficiency is necessary since RNAi was effective at increasing RNAi.

Response 5: While RNAi was indeed successful at knocking down nuclease function, we demonstrated that under some conditions, such as injection or sequential feeding, *yellow*-dsRNA alone was insufficient to achieve transcript knockdown (Figure 3), indicating that RNAi was not effective until the nuclease function was ablated. We have added text in the Discussion (lines 385-397) to highlight that previous attempts at sterilization of males fell short of requirements for field deployment (Cruz et al. 2018).

6) Conversely, it is unclear if the new level of RNAi efficiency would be sufficient to achieve the goals of the researchers/field, e.g., to produce male sterility." Please address these two paragraphs as well in the point by point response. Thanks.

Response 6: In SIT programmes for tephritids, male sterility rates of >99% are demanded; a new reference has been added in the Discussion highlighting this point (FAO/IAEA/USDA 2014; line 392). If RNAi-based sterilization is to serve as a possible replacement of radiation-based methods, the RNAi efficacy would need to be improved from what we previously achieved with this species using an RNAi-mediated knockdown of spermatogenesis genes. We have now added a comment to the Discussion on this point too (lines 385-397). Based on our ability to almost completely eliminate yellow activity using RNAi in combination with knockdown of nucleases (based on the measurements of melanisation assays), we believe that nuclease knockdown might provide the added degree of knockdown needed for SIT applications. Please see lines 385-397 of the Discussion. As we did not wish to focus only on field applications, we also wanted to highlight that improved RNAi efficacy may be very helpful for basic science research as well, to explore gene functions through more effective knockdown techniques.

Additionally, here are the line numbers of other edits made:

tryoni (line 31)
haemocoel (lines 44; 63; 263)
DsRNA (line 118; 146)
To amplify a 347 bp fragment (line 122)
six days later (line 153)
gfp- or *yellow*- dsRNA (line 156)
qRT-PCR (line 167)
containing (line 168)
cDNA (line 207)
technical replicates (line 227)
male and female (line 252)
day 3 (line 301)
Equally important (line 375)

Please refer to lines 218-224 for the edits to the manuscript that address the concern about the use of 1 reference gene.

Lines 292-293; 368-374 address the concern about using a non-specific dsRNA control.

We have clarified the doses applied in all treatments for the sequential feeding assays (lines 156-184).

The grade and purity of dsRNA purchased from AgroRNA is included in line 120.

Centrifugation of gut contents and minimization of intracellular nucleases (lines 137-138; 259-260).

We have added a comment in the results section highlighting the differences between the two nucleases and the reason why two phylogenetic trees were necessary (lines 249-250).

Figure 3 caption revised for clarification.

Figure captions revised to include dsRNA doses used in each treatment and to include the correct spelling of “dsRNase”.

We have added 3 new references (reference #16; lines 59; 226), (reference #45; line 373), (reference # 45; line 394)